# Curvature-induced expulsion of actomyosin bundles during cytokinetic ring contraction

Junqi Huang[1]\*[†], Ting Gang Chew[1]\*[†], Ying Gu[2][‡], Saravanan Palani[1][‡], Anton Kamnev[1], Douglas S Martin[3], Nicholas J Carter[1], Robert Anthony Cross[1], Snezhana Oliferenko[2][§], Mohan K Balasubramanian[1]\*

[1]Division of Biomedical Sciences, Warwick Medical School, University of Warwick, Coventry, United Kingdom; [2]Randall Division of Cell and Molecular Biophysics, King's College London, London, United Kingdom; [3]Department of Physics, Lawrence University, Appleton, United States

\*For correspondence: junqi. huang@warwick.ac.uk (JH); t.g. chew@warwick.ac.uk (TGC); m.k. balasubramanian@warwick.ac.uk (MKB)

[†]These authors contributed equally to this work
[‡]These authors also contributed equally to this work

Present address: [§]Francis Crick Institute, London, United Kingdom

**Abstract** Many eukaryotes assemble a ring-shaped actomyosin network that contracts to drive cytokinesis. Unlike actomyosin in sarcomeres, which cycles through contraction and relaxation, the cytokinetic ring disassembles during contraction through an unknown mechanism. Here we find in *Schizosaccharomyces japonicus* and *Schizosaccharomyces pombe* that, during actomyosin ring contraction, actin filaments associated with actomyosin rings are expelled as micron-scale bundles containing multiple actomyosin ring proteins. Using functional isolated actomyosin rings we show that expulsion of actin bundles does not require continuous presence of cytoplasm. Strikingly, mechanical compression of actomyosin rings results in expulsion of bundles predominantly at regions of high curvature. Our work unprecedentedly reveals that the increased curvature of the ring itself promotes its disassembly. It is likely that such a curvature-induced mechanism may operate in disassembly of other contractile networks.

## Introduction

Many eukaryotes assemble a dynamic actomyosin-based contractile ring to divide one cell into two (*Pollard and Wu, 2010*). During ring contraction, ring components continuously disassemble while maintaining ring contractility (*Pollard and Wu, 2010*). This is in stark contrast to other actomyosin machineries, particularly the thick and thin filament based sarcomeres in muscle fibers, which undergo contraction and relaxation cycles without disassembly (*Murrell et al., 2015*). ADF1/Cofilin together with Coronin and WD-repeat protein Aip1 modulate actin turnover by severing actin bundles in many types of actin networks (*Jansen et al., 2015*). However, their involvement in actomyosin ring contraction and disassembly is still unclear (*Chen and Pollard, 2011*). The F-actin based molecular motor Myosin II may also regulate disassembly of actin networks during cytokinesis (*Guha et al., 2005*; *Murthy and Wadsworth, 2005*), presumably by myosin II mediated actin filaments buckling and breakage (*Murrell and Gardel, 2012*; *Reymann et al., 2012*; *Vogel et al., 2013*). The exact mechanism governing disassembly of actomyosin rings and whether and how the ring disassembly is coordinated to ring contraction have remained unknown so far.

To address these gaps in our knowledge, we studied the dynamics of actomyosin ring contraction and disassembly in the fission yeasts *Schizosaccharomyces japonicus* and *Schizosaccharomyces pombe*. We report that micron-scale actin bundles are expelled during contraction of the actomyosin ring in a ring curvature-dependent manner in intact cells, in spheroplasts, and when isolated

actomyosin rings are triggered to contract with ATP. We test the curvature-dependence of the actomyosin ring disassembly by physically deforming protoplasts to defined geometries.

## Results and discussion

Whilst studying F-actin dynamics in *S. japonicus* cells expressing LifeAct-GFP, as a marker for all actin structures (*Gu et al., 2015*; *Huang et al., 2012*; *Riedl et al., 2008*), large F-actin bundles appeared to be connected with the contracting ring (*Figure 1A*). We imaged actin rings in the axial plane to achieve higher resolution by preparing spheroplasts, in which actomyosin rings were oriented randomly and slid along the cell membrane during contraction, consistent with previous work in *S. pombe* and in reconstituted giant unilamellar vesicles (GUVs) (*Mishra et al., 2012*; *Miyazaki et al., 2015*). In spheroplasts where the actomyosin ring was positioned close to the imaging plane, actin filaments appeared to project outwards from the contracting ring (*Figure 1B*). Permeabilized spheroplasts retained substantial numbers of actin filaments that remained attached to the ring, indicating that the actin filaments were connected to the contracting ring (*Figure 1C*). To ensure that the observed actin filaments were not an artifact of LifeAct-GFP used in these experiments, we fixed and stained wild type cells (that did not express any fusion protein) and wild type cells expressing LifeAct-GFP with Rhodamine-conjugated phalloidin (*Figure 1—figure supplement 1A*). We also fixed and stained permeabilized spheroplasts generated from these two strains with Rhodamine-conjugated phalloidin. In both cases, we were able to detect actin filaments connected to the contracting actomyosin ring (*Figure 1—figure supplement 1A*), ruling out the possibility that the observed filaments were an artifact of the tag used. When fixed cells expressing Rlc1p-GFP were

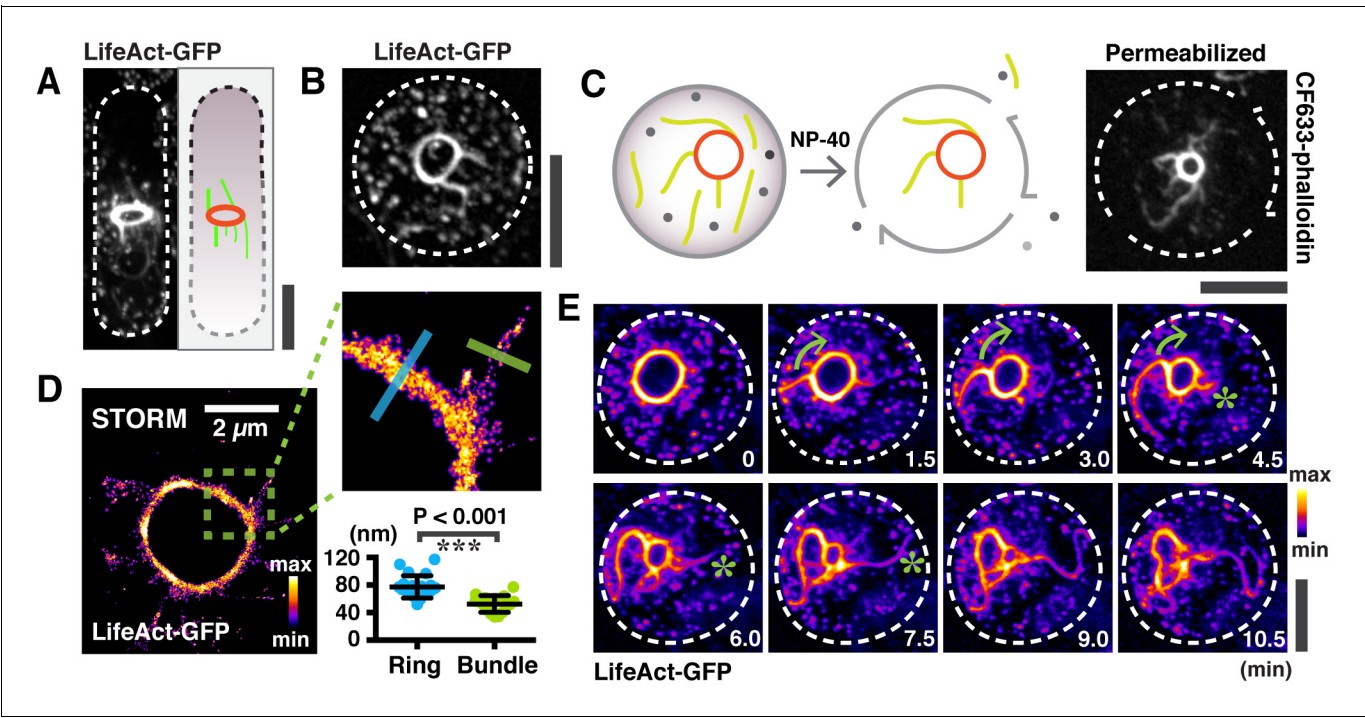

**Figure 1.** Large actin bundles are expelled during actomyosin ring contraction. (**A**) *S. japonicus* cells expressing LifeAct-GFP. (**B**) Spheroplasts expressing LifeAct-GFP. (**C**) Permeabilized spheroplasts (by 0.5% NP-40) stained with CF633-phalloidin. (**D**) Quantification of the widths of actin rings and bundles by STORM. n = 16 random bundles and 21 random positions of 5 rings. The full-width-half-maximum (FWHM) value of a Gaussian-fitted line was measured to represent the widths. (**E**) Time-lapse micrographs of spheroplasts expressing LifeAct-GFP. Asterisks: expelling actin bundles. Arrows: rotations of an actin bundle. All scale bars are 5 μm unless specified. Error bars: s.d.

The following figure supplement is available for figure 1:

**Figure supplement 1.** Analysis of actin bundles during actomyosin ring contraction and disassembly in *S. japonicus and S. pombe*.

stained with Rhodamine-conjugated phalloidin, actin filaments attached to the ring colocalized with Rlc1p, suggesting that Rlc1p-GFP (and by extension Myosin II) was also part of the actin filaments attached to contracting rings (*Figure 1—figure supplement 1A*). Furthermore, actin filaments linked to the contracting actomyosin ring were detected in *S. pombe* cells and permeabilized spheroplasts of wild type cells (with no genetic tag) and of wild type cells expressing LifeAct-GFP or Rlc1-GFP (*Figure 1—figure supplement 1B*), establishing that this phenomenon is not unique to *S. japonicus* cells and spheroplasts. Actin and myosin II filaments attached to contracting actomyosin rings have been observed by Wollrab and colleagues in *S. pombe* cells, although the significance of these structures was not investigated in their work (*Wollrab et al., 2016*). In the rest of the manuscript, unless specifically indicated, experiments were performed in *S. japonicus*.

Analysis of actomyosin rings with super resolution microscopy STORM showed it to be 77 ± 16 nm wide for ring diameters ranging 500 nm to 2.8 µm, whilst the ring associated actin filamentous structures were 52 ± 12 nm wide (*Figure 1D* and *Figure 1—figure supplement 1C*; enlarged panel), indicating that these structures contained multiple 7 nm-wide actin filaments arranged in a bundle (*Takaine and Mabuchi, 2007*). On this basis, we now refer to them as ring-associated actin bundles.

Time-lapse microscopic analysis of spheroplasts expressing LifeAct-GFP showed that actin bundles appeared to emerge from the contracting ring, continued to extend in length and were progressively expelled from the sites of their initial appearances (*Figure 1E*; *Video 1*). Occasionally, upon the expulsion of actin bundles, the contracting ring seemed to disintegrate at the region of release (*Figure 1—figure supplement 1D*). Similar observations of actin bundles being expelled from contracting rings were also made in *S. pombe* (*Figure 1—figure supplement 1E*). Actin bundles were also expelled during contraction in *S. japonicus* intact cells expressing LifeAct-GFP in which the GFP intensity was reduced and actin appeared disorganized near the end of cytokinesis (*Figure 1—figure supplement 1F and G*). The actin intensity measured in permeabilized spheroplasts using CF633-phalloidin staining in large rings was higher than in small rings (*Figure 1—figure supplement 1H*; Note that the fluorescence intensity from ring-associated bundles were not considered in this analysis). However, the combined actin fluorescence intensity from rings and associated bundles was roughly comparable in large and small rings (*Figure 1—figure supplement 1I*), suggesting that the expulsion of actin bundles from the ring may represent a mechanism for ring disassembly. To estimate the fraction of ring associated actin filaments that were expelled as bundles during contraction, we measured the fluorescence intensity of the bundle fraction in contracting rings using LifeAct-GFP as a marker (*Figure 1—figure supplement 1J*). We estimated that ~68% of actin filaments that are lost from the ring during contraction (from diameters of 3.48 ± 1.44 µm to 1.23 ± 0.65 µm) are expelled as bundles.

Other components of actomyosin rings such as the regulatory light chain of myosin Rlc1p, tropomyosin Cdc8p (co-staining of these shown in *Figure 2A*), IQGAP protein Rng2p, and F-BAR protein Cdc15p (*Figure 2A*) were present in actin bundles associated with contracting rings in spheroplasts. We investigated the dynamics of ring contraction using each of these proteins as a marker and found that GFP-Rng2p, Cdc15p-GFP, and Rlc1p-GFP were associated with the expelling/expelled bundles (*Figure 2B*, *Video 2*). In time-lapse imaging experiments, Rlc1p-GFP was also found in bundles associated with contracting rings in *S. pombe* (*Figure 2—figure supplement 1A*). When the contracting ring was analyzed in spheroplasts of *S. japonicus*, actomyosin bundles became detectable when the ring contracted to a diameter of ~2 µm and were expelled from the ring at the rate of 2.19 ± 0.61 µm/min, (*Figure 2C,D*; *Figure 2—figure supplement 1B*).

Profiling the distribution of ring-associated bundles in spheroplasts indicated that there was

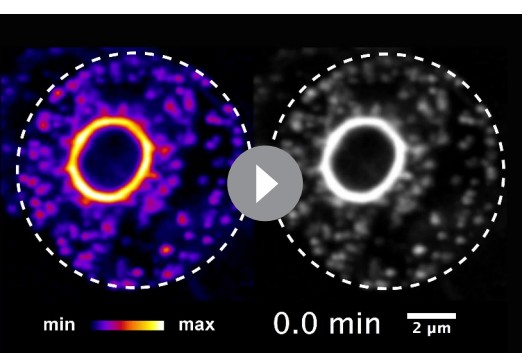

**Video 1.** Expulsion of F-actin bundles during ring contraction in a spheroplast expressing LifeAct-GFP. Asterisks mark the elongation of an actin bundle. Left panel is presented in 'Fire' lookup table and its calibration bar. Spheroplasts are outlined by dotted lines. Time zero indicates the start of the video.

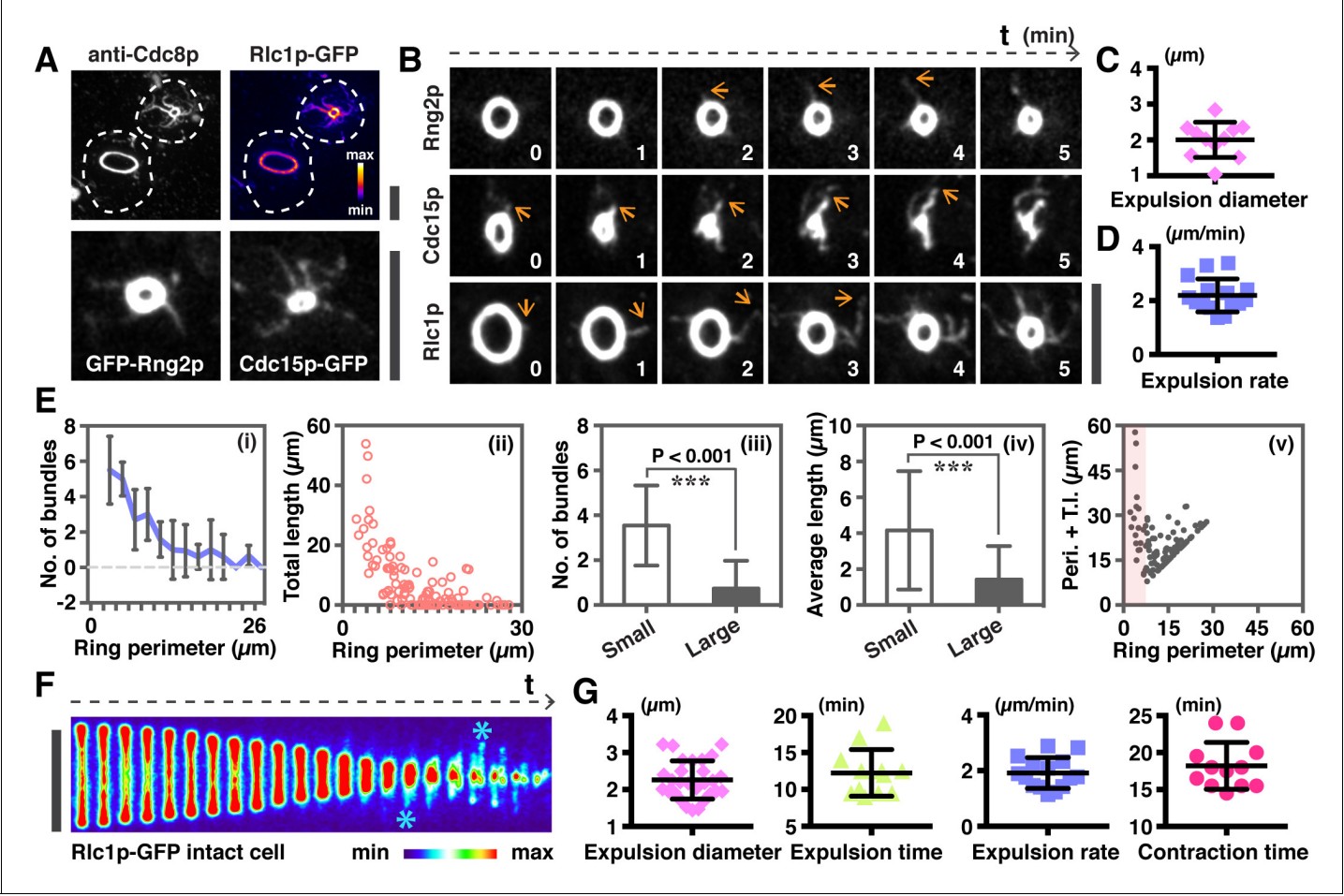

**Figure 2.** Quantitative analysis of actomyosin bundle expulsion. (**A**) Top panel: localizations of tropomyosin Cdc8p and Rlc1p-GFP in permeabilized spheroplasts. Both proteins were detected by immunofluorescence microscopy. Bottom panel: spheroplasts expressing GFP-Rng2p, and Cdc15p-GFP were imaged to show the localization of these proteins on ring-associated bundles. (**B**) Time-lapse micrographs of spheroplasts expressing GFP-Rng2p, Cdc15p-GFP, and Rlc1p-GFP, respectively. The time is indicated at bottom right corner in minutes. (**C**) Quantification of the Rlc1p-GFP ring diameter during initial bundle expulsion (n = 11 rings) in spheroplasts. The ring diameters were calculated by fitting the measured perimeters in *Figure 2—figure supplement 1B* with Diameter = Perimeter / π. (**D**) Quantification of the Rlc1p-GFP bundle-released rate (n = 15 bundles) in spheroplasts. (**E**) Quantitative analysis of bundle expulsion in permeabilized spheroplasts. i: quantification of the number of bundles as a function of ring perimeter; ii: total bundle lengths as a function of ring perimeter; iii: number of bundles associated with small or large rings; iv: average bundle length associated with small or large rings; v: sum of total bundle length (T.l.) and ring perimeter (Peri.) as a function of ring perimeter. Large: ring perimeter ≥ 11 μm; Small: ring perimeter < 11 μm; n = 75 (Large), 46 (Small), respectively. Graphs ii and v: each data point represents one permeabilized spheroplast, n = 121. (**F**) Kymographs of contracting rings at 1 min interval in intact cells expressing Rlc1p-GFP. Asterisks: actomyosin bundles. (**G**) Quantifications of the ring diameter during initial bundle expulsion (n = 28 rings), time of bundle expulsion (n = 12 rings), traveling rate of expelled bundles (n = 15 bundles), and total ring contraction time (n = 12 rings) in intact cells. Scale bar: 5 μm. Error bars: s.d.

The following figure supplement is available for figure 2:

**Figure supplement 1.** Analysis of actomyosin ring components during ring contraction and disassembly.

an inverse correlation (*Figure 2Ei*) between the number of actin bundles (measured using tropomyosin Cdc8p as a marker for actin bundles) or bundle lengths with the perimeter of the actomyosin ring (*Figure 2Eii*). It appeared that there were on average 3.5 ± 1.8 actin bundles attached to rings of less than 11 μm perimeter (corresponding to about 3.5 μm ring diameter), whereas there were 0.7 ± 1.2 actin bundles associated to rings of larger perimeter (*Figure 2Eiii*). When the length of individual ring-associated actin bundles was measured, we found that on average the bundle was longer in small rings (4.16 ± 3.30 μm for perimeter of <11 μm) than in larger rings (1.42 ± 1.86 μm for

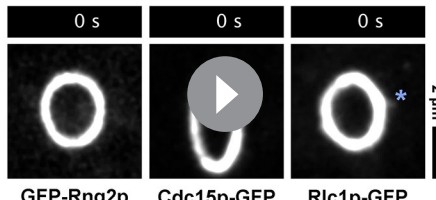

**Video 2.** Expelled filaments/bundles contain GFP-Rng2p, Cdc15p-GFP, and Rlc1p-GFP. Asterisks indicate expelling bundles. Time zero indicates the start of the video.

perimeter of >11 μm) (*Figure 2Eiv*). The total length of ring perimeters and actin bundles associated with the ring was higher in smaller rings compared to larger rings (*Figure 2Ev*). These analyses indicated that ring disassembly and bundle expulsion was coupled to ring contraction.

In mitotic cells expressing Rlc1p-GFP (*Figure 2F*, *Figure 2—figure supplement 1C*) or GFP-Rng2p (*Figure 2—figure supplement 1D*), actomyosin bundles were also not detected in rings with a larger diameter. The bundles associated with the actomyosin ring became more prominent when the actomyosin ring contracted to a diameter of 2.26 ± 0.52 μm (*Figure 2G*; *Figure 2—figure supplement 1E*; *Video 3*; initial

ring diameter was 5.49 ± 0.37 μm). On average, the ring-associated bundles became detectable after 12.3 ± 3.2 min of initiation of actomyosin ring contraction, and continued to be expelled from the actomyosin ring until the end of contraction at a rate of 1.91 ± 0.55 μm/min (*Figure 2G*; full contraction time was 18.21 ± 3.18 min). Hence, it appeared that the expulsion of actomyosin bundles became obvious when rings contracted to ~40% of the initial diameter.

We then asked if cytoplasmic factors were required for the expulsion of actin bundles during ring contraction (*Figure 3A*). To investigate this, we established a *S. japonicus* cell ghost system (depleted of cytosol) which contained contractile actomyosin rings, as previously described in *S. pombe* and *S. cerevisiae* (*Mishra et al., 2013*; *Young et al., 2010*). Treating *S. japonicus* cell ghosts with ATP induced contraction of isolated actomyosin rings (*Figure 3—figure supplement 1A*). Interestingly, the actin bundles remained associated with the contracting and fully contracted rings (*Figure 3B*, *Video 4*) although the F-actin signal as visualized with LifeAct-GFP was lesser than in cells and spheroplasts. To quantitate aspects of actin bundle expulsion in cell ghosts, we fixed and stained contracting and contracted actomyosin rings with CF633-phalloidin (*Figure 3C*). The ring-associated bundles appearing in cell ghosts after ATP addition were similar to those observed during in vivo ring contraction (*Figure 2E*). We observed 4.3 ± 1.4 bundles associated with the rings in cell ghosts, with individual average length of 3.9 ± 1.3 μm (*Figure 3D*; total bundle length 16.1 ± 6.7 μm). The sum F-actin intensity of cell ghosts (ring actin + bundle actin) following ATP addition showed no significant changes, whereas there was a decrease of actin intensity in the rings (*Figure 3E*). We estimated protein levels of actin and the actin filament binding protein tropomyosin Cdc8p in the supernatant and pellet fractions following centrifugation of contracting actomyosin rings in cell ghosts (treated with ATP for 20 or 40 min). We did not find a significant increase in the amount of actin and Cdc8p in the supernatant in cell ghosts treated with ATP for 20 or 40 min

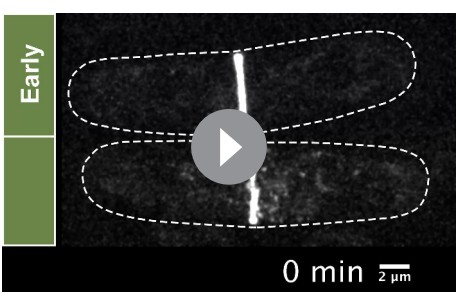

**Video 3.** Ring contraction in intact *S. japonicus* cells expressing Rlc1p-GFP. Asterisks indicate actomyosin bundles are expelled from the contracting rings. Time zero indicates the start of the video.

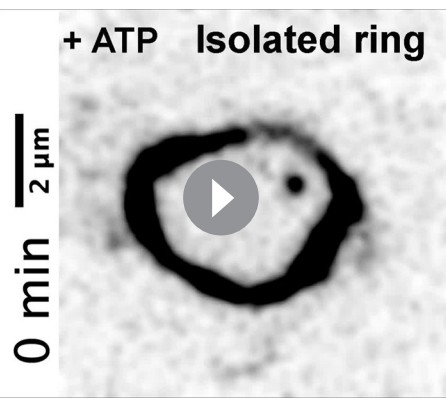

**Video 4.** Expulsion of actin bundles in 0.5 mM ATP-treated isolated rings in cell ghosts. Asterisks indicate expelled actomyosin bundles during contraction. Time zero indicates the start of the video.

(*Figure 3F*), consistent with the idea that ring contraction resulted in denser actin filaments containing binding proteins such as tropomyosin that remain in the pellet fraction. These results indicated that in cell ghosts treated with ATP for 20 to 40 min, the rings reduced in actin amounts. However, this did not lead to an increase in soluble G-actin during ring contraction, suggesting that most actin was expelled in filaments contained in bundles during this period of time. Furthermore, these experiments established that the expulsion of actin bundles was a result of ring contraction and was independent of continuous presence of cytosol. Finally, the absence of cytosolic components such as G-actin also suggests that the actomyosin bundles are expelled as a result of contraction rather than the possibility that these bundles were newly polymerized actin cables that were not incorporated into the highly curved actomyosin ring (*Figure 3A*).

What is the mechanism of expulsion of actin bundles during ring contraction? We had previously noted that actomyosin bundles were expelled at later stages of contraction. As the actomyosin bundles were expelled from contracting rings of smaller diameters, which have higher curvatures, we speculated that the local curvature of rings might induce bundle expulsion. In spheroplasts expressing Rlc1p-GFP, the expelled bundles became visible and their numbers increased with the progression of time and increasing the curvature (*Figure 4A*). Although rings started with curvatures (1/radius) of even smaller than 0.4 $\mu m^{-1}$ (translates to a ring diameter of >5 $\mu m$ in

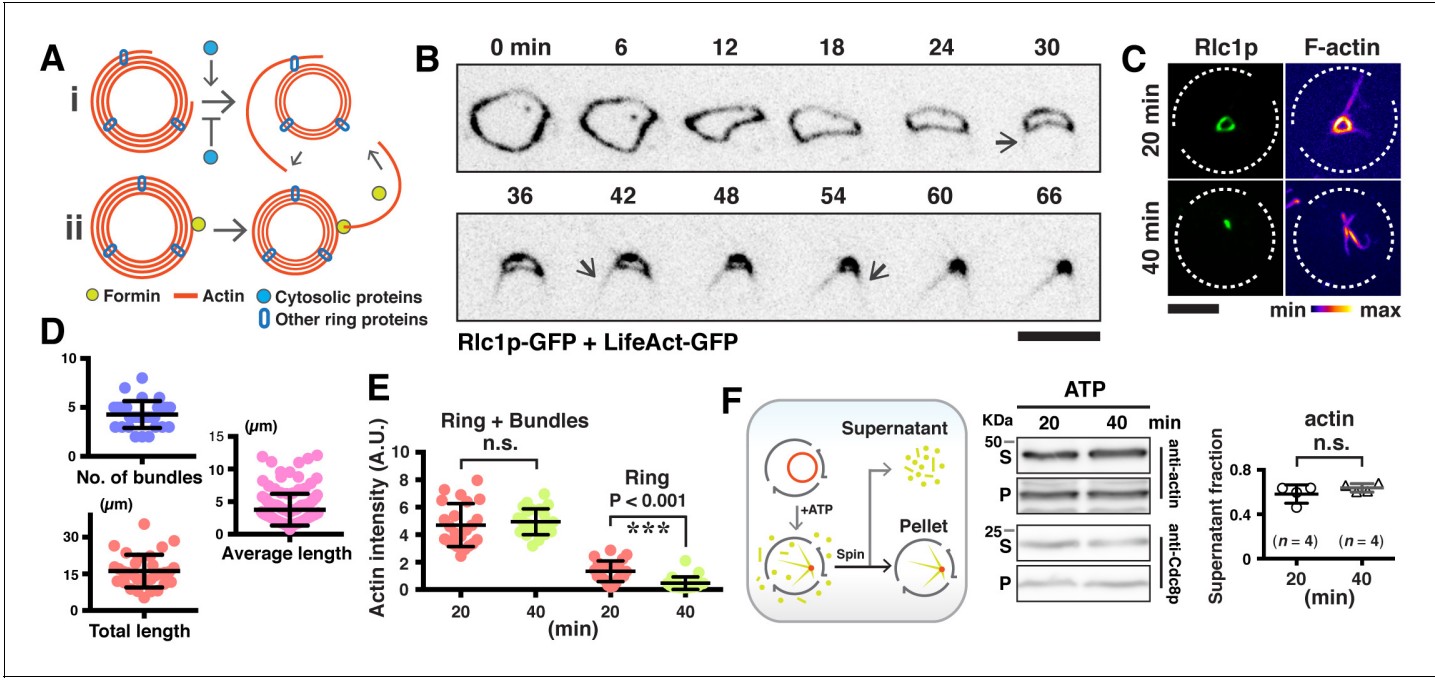

**Figure 3.** Expulsion of actomyosin bundles during contraction of isolated rings. (**A**) i, bundle expulsion mediated by cytosolic proteins; ii, newly polymerized actin filaments that are unable to incorporate into a highly curved actomyosin ring. (**B**) Time-lapse micrograph shows contraction of an actomyosin ring in a cell ghost after addition of LifeAct-GFP (concentration: ~4 ng/μl) and 0.5 mM ATP. Arrows: expulsion of actomyosin bundles. (**C**) Cell ghosts containing Rlc1p-GFP were treated with 0.5 mM ATP for 20 min or 40 min respectively, and stained with CF633-phalloidin to visualize F-actin structures. (**D**) Quantification of the number of expelled bundles, total bundle length, and average bundle length in each cell ghost after 40 min ATP treatment (n = 32 cell ghosts, 137 bundles). (**E**) Quantification of the sum actin intensity in cell ghosts (ring + bundles) or ring (ring) after treatment with ATP. To measure the total actin intensity in a cell ghost stained with CF633-phalloidin, a square to cover the actomyosin ring and the associated bundles as the region-of-interest was selected, and the sum intensity of the region-of-interest was measured. For the actin intensity of the ring in cell ghosts, a line along the ring circumference to cover the ring area was selected as the region-of-interest for sum intensity measurement. n = 23, 24 cell ghosts for 20 min, 40 min time point, respectively. ns: no significant difference. ***p<0.001. (**F**) Immunoblots of actin and Cdc8p from cell ghosts treated with ATP for 20 min and 40 min. S, Supernatant: soluble G-actin or short F-actin. P, Pellet: actomyosin rings and associated bundles. Four blots were used for quantifications. Scale bar: 5 μm. Error bars: s.d.

The following figure supplement is available for figure 3:

**Figure supplement 1.** Contraction of the isolated Rlc1p-GFP ring in cell ghosts upon ATP treatment.

spheroplasts), the mean curvature at which bundles began to be expelled was 1.1 ± 0.4 μm$^{-1}$ (translates to a ring diameter of ~2 μm) (**Figure 4A**). Given that bundle expulsion was only observed after a significant reduction in ring diameter, suggesting that bundle expulsion in rings is promoted by increased curvature. Furthermore, even in partially contracted and curved actomyosin rings in ATP

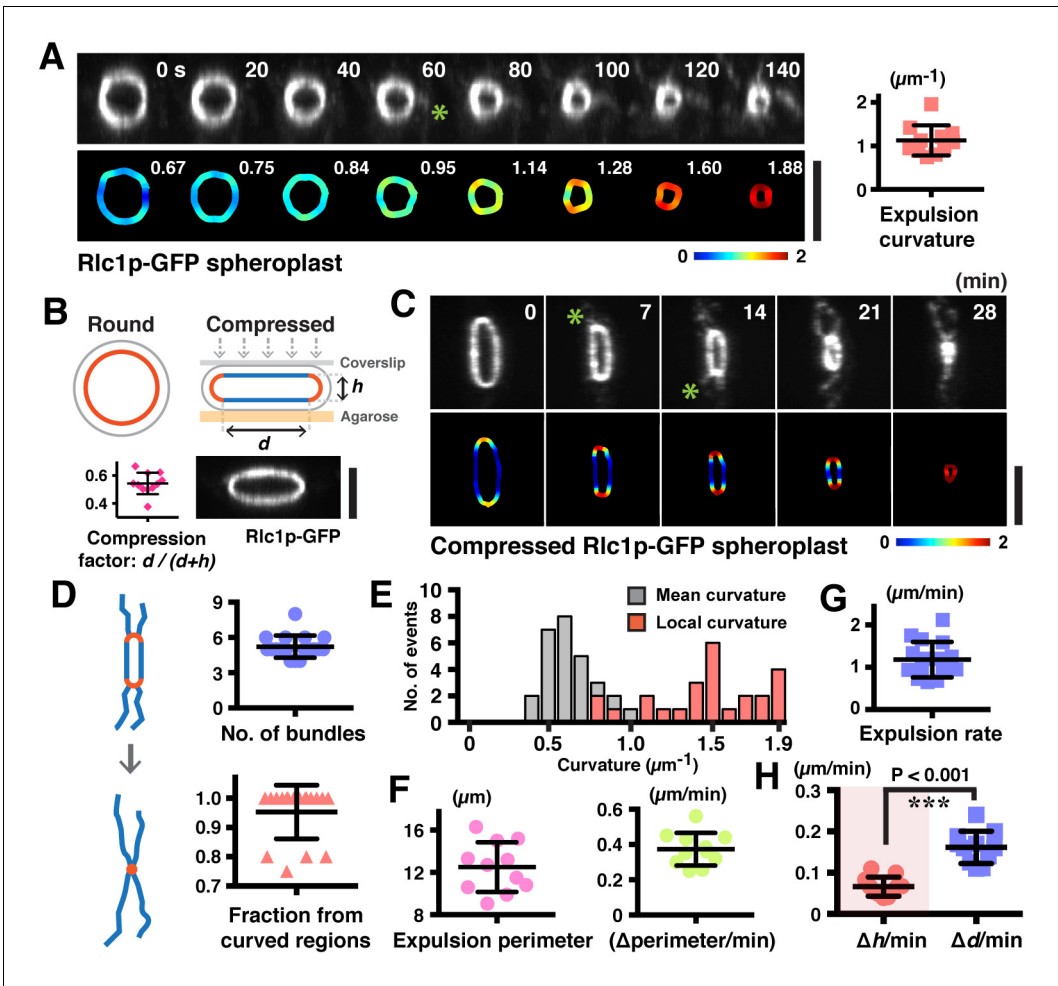

**Figure 4.** Increased ring curvature promotes actomyosin bundle expulsion. (**A**) Analysis of the curvature of sliding rings in spheroplasts expressing Rlc1p-GFP (n = 11 rings). The ring curvatures were color-coded, and mean curvatures of each stage of contraction are shown in the lower panel. The mean ring curvature during initial bundle expulsion was plotted in the right graph. Asterisk: an expelled bundle. (**B**) Distorted actomyosin rings after spheroplasts were compressed mechanically (n = 11 rings). *d*: flat; *h*: curved regions. The degree of compression was quantified by compression factors. (**C**) Micrographs of bundle expulsion from the curved regions of rings. The ring curvatures are color-coded. Asterisks: expelled bundles. (**D**) Quantification of the number of ring-associated bundles, and fraction of bundles expelled from curved regions (n = 94 bundles from 18 rings). (**E**) Quantification of the local curvatures at the initial sites of bundle expulsion and the mean ring curvatures during bundle expulsion (n = 25 bundles from 11 rings). (**F**) Quantification of the ring perimeter during initial bundle expulsion and the rate of perimeter change in contracting rings (n = 11 rings). (**G**) Quantification of the traveling rate of expelled bundles (n = 16 bundles) (**H**) Rates of contraction in flat (△*d*) and curved (△*h*) regions (n = 11 rings). Scale bar: 5 μm. Error bars: s.d.

The following figure supplement is available for figure 4:

**Figure supplement 1.** Curvature analysis of actin rings in cell ghosts and Rlc1p rings after compression of spheroplasts.

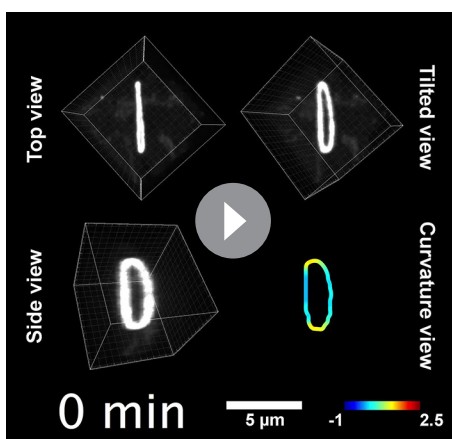

**Video 5.** Curvature-induced expulsion of actomyosin bundles in distorted rings. Curvature of distorted rings is color-coded. Ring contraction is shown in three different angles of projections. Time zero indicates the start of the video.

treated cell ghosts, the majority (~68%) of ring-associated bundles were detected at regions of increased local curvature where the ring appeared to be bent (*Figure 4—figure supplement 1A*).

To directly test the hypothesis that the ring disassembly by bundle expulsion depends on ring curvature, we compressed the spheroplasts expressing Rlc1p-GFP to artificially produce rings with distinctive curved and flat regions (*Figure 4B*; compression factor ~0.54 ± 0.08). Strikingly, about 95% of the actomyosin bundles were expelled from the curved regions of compressed rings compared to flat regions (*Figure 4C,D*; *Figure 4—figure supplement 1B* and *Video 5*; average 5.2 ± 0.9 bundles per ring), and the local curvature at the sites of expelled bundles was 1.5 ± 0.3 $\mu m^{-1}$ (*Figure 4E*; mean curvature of compressed rings was 0.7 ± 0.2 $\mu m^{-1}$). Whereas actomyosin bundles were expelled when uncompressed rings contracted to a perimeter of 6.3 ± 1.5 $\mu m$ (*Figure 2—figure supplement 1B*), majority of compressed rings started releasing actomyosin bundles when contracted to a perimeter of only 12.5 ± 2.4 $\mu m$, indicating that the curvature change induced expulsion of actomyosin bundles (*Figure 4F*). Curiously, following compression, the ring contraction rate ($\triangle$perimeters/time) was 0.37 ± 0.09 $\mu m$/min, which was significantly less than the 1.26 ± 0.57 $\mu m$/min observed in uncompressed spheroplasts (*Figure 4F* and *Figure 2—figure supplement 1B*). The bundles were expelled at a reduced rate of 1.18 ± 0.42 $\mu m$/min as compared to 2.19 ± 0.61 $\mu m$/min in uncompressed spheroplasts (*Figure 4G* and *Figure 2D*). Furthermore, shortening of the flat regions ($\triangle d$) of compressed rings occurred faster than the decrease of ring heights ($\triangle h$) (*Figure 4H*; $\triangle d$/min= 0.16 ± 0.04 $\mu m$/min vs. $\triangle h$/min = 0.07 ± 0.02 $\mu m$/min). Thus, the actomyosin bundles were constrained by the local curvature of contracting rings, and their expulsion depends on the curvature changes during ring contraction. Our results showed that ring disassembly is coupled physically to ring contraction in a ring-geometry coordinated manner.

The expulsion of micron-scale actomyosin bundles may serve as a major mechanism to disassemble actomyosin rings during cytokinesis (*Figure 4—figure supplement 1C*). It is presently unclear if other mechanisms, such as cofilin and myosin II mediated filament severing / breakage and loss of actin monomers also contribute to actin disassembly during cytokinesis. New probes exhibiting reduced photobleaching and directly reporting actin cytoskeleton dynamics will be required to help estimate if other mechanisms are in operation as well. The coupling of bundle expulsion mediated-ring disassembly to ring contraction appears intrinsic to the ring itself, and is potentially driven by a physical mechanism that is independent of cytosolic factors. It is possible that the affinity of binding of unidentified actin binding proteins with F-actin is reduced upon increased curvature of actin filaments. Thus, actomyosin bundles may not be able to bend sufficiently to accommodate the high ring curvature at later times in contraction, leading to their expulsion from the ring. It is likely that attainment of a curved morphology is an energetically unfavorable conformation since actin filaments have a persistence length of ~10 microns (which can increase further with bundling) (*Ott et al., 1993*; *Riveline et al., 1997*).

Although we were able to visualize actin filaments, Rlc1p-GFP, GFP-Rng2p, and Cdc15p-GFP on expelled bundles in living cells and spheroplasts, we were only able to visualize expulsion of actin filaments in bundles in contracting actomyosin rings within cell ghosts (*Figure 3—figure supplement 1A* and data not shown for GFP-Rng2). It is possible that other ring proteins do not associate tightly to actin bundles during ring contraction in a less crowded environment (such as permeabilized cell ghosts), whereas they are able to associate with expelled bundles in a cellular environment of increased molecular crowding. Alternatively, it is possible that, in the presence of cytosol, as in a

cellular context (*Pelham and Chang, 2002*; *Wong et al., 2002*), loss of ring proteins from actin bundles in the ring is balanced by replenishment from the cytosolic pool. However, in cell ghosts, such replenishment of ring proteins does not occur due to the absence of cytosol. Future work should investigate these possibilities.

Several types of actin filament behaviors such as branching and bundling are coupled to the curvature of the actin filament network (*Liu et al., 2008*; *Risca et al., 2012*). Expulsion of the actomyosin bundles that occurs when the cytokinetic actomyosin ring acquires high curvature suggests that curvature-induced disassembly could be a general mechanism for disassembly of actomyosin networks.

## Materials and methods

### Yeast strains, medium, and culture conditions

*S. japonicus* and *S. pombe* strains used in this study are listed in Table S1 (*Supplementary file 1*). *S. japonicus* strains were constructed as previously described (*Gu et al., 2015*). *S. japonicus* and *S. pombe* cells were cultured in YEA medium (5 g/l yeast extract, 30 g/l glucose, 225 mg/l adenine) at 24°C until mid-log phase for physiological analysis.

### *S. japonicus* spheroplasts and cell ghosts preparation, and ATP-dependent contraction of isolated rings

Isolated actomyosin rings of *S. japonicus* in cell ghosts were prepared using similar published methods (*Huang et al., 2016*; *Mishra et al., 2013*). Briefly, *S. japonicus* cells were cultured in YEA medium overnight to mid-log phase, and 15 ml of cells were spun down and washed once with an equal volume of SCS buffer (20 mM sodium citrate [pH 5.8], 1 M sorbitol). Cells were then digested in 1 ml SCS buffer containing 0.1 g/ml lytic enzymes Lallzyme MMX (Lallemand) (*Flor-Parra et al., 2014*). The resulting spheroplasts were then washed and recovered in YEA medium containing 1 M sorbitol until the actomyosin rings were assembled. Spheroplasts were then washed once with wash buffer (20 mM PIPES–NaOH [pH 7.0], 0.8 M sorbitol, 2 mM EGTA, 5 mM $MgCl_2$), and then permeabilized with isolation buffer (50 mM PIPES–NaOH [pH 7.0], 0.16 M sucrose, 50 mM EGTA, 5 mM $MgCl_2$, 50 mM potassium acetate) containing 0.5% Nonidet P40 (NP-40) to obtain cell ghosts containing isolated rings. The suspension was washed twice with reactivation buffer (0.16 M sucrose, 5 mM $MgCl_2$, 50 mM potassium acetate, 20 mM MOPS–NaOH [pH 7.0], pH adjusted to 7.5). To induce ATP-dependent actomyosin ring contraction, the cell ghosts were treated with reactivation buffer containing 0.5 mM ATP (Sigma; A6559).

### Sample preparation for microscopy imaging

To image cells in suspensions, 1 ml of a yeast culture at mid-log growth phase was first spun down at 450 xg for 2 min, and ~16 µl of concentrated samples were loaded onto an Ibidi µ-Slide 8-Well glass bottom dish (Cat. No. 80827). To image spheroplasts in suspension, 1 ml of spheroplasts (regenerated in YEA medium containing 1 M sorbitol for 1.5 to 3 hr) was spun down at 450 xg for 2 min and ~16 µl of the concentrated suspension was loaded onto an Ibidi µ-Slide 8-Well glass bottom dish. To image contraction of rings in cell ghosts, equal volume of 1 mM ATP (final concentration 0.5 mM ATP) was added to cell ghosts and imaged in an Ibidi µ-Slide 8-Well glass bottom dish. All imaging dishes were sealed with an adhesive film membrane or mineral oil to prevent evaporation during imaging.

To generate actomyosin rings of distorted shapes in *Figure 4B,C* and *Figure 4—figure supplement 1B*, spheroplasts were prepared as described above, and were compressed by capillary forces pulling the coverslip and slide containing an agarose pad together. The slide was then sealed with VALAP prior to imaging. All samples were imaged at room temperature.

The CellASIC ONIX Microfluidic system was used to immobilize cells for imaging in *Figure 1—figure supplement 1F and G*, *Figure 2F,G*, and *Figure 2—figure supplement 1C and E*.

To estimate the fraction of actin filaments that were expelled as bundles from the contracting rings in *Figure 1—figure supplement 1J*, two Z-stacks (0.2 µm step size) of the same spheroplasts expressing LifeAct-GFP were acquired consecutively at a 12 min interval to minimize photobleaching. To quantitate the effect of photobleaching during imaging, the neighboring spheroplasts in the

imaging field were selected and measured for their sum fluorescence intensity of LifeAct-GFP between two time points. The ratio of the sum fluorescence intensity of late versus early time points of selected spheroplasts was 1.018 ± 0.044, indicating that the photobleaching effect was minimal. Only spheroplasts without obvious actin bundles at the beginning of imaging (average ring diameter is 3.48 ± 1.44 µm) were chosen.

## Immunofluorescence microscopy and F-actin staining

Spheroplasts were first permeabilized with isolation buffer containing 0.5% NP-40, and washed twice with reactivation buffer. Permeabilized spheroplasts were then fixed with 3.7% formaldehyde for 12 min at room temperature and washed twice with reactivation buffer. A rabbit primary antibody recognizing *S. pombe* tropomyosin Cdc8p (*Balasubramanian et al., 1992*) and a mouse CF633 anti-GFP (Sigma; SAB4600146) were added at 1:400 and 1:200, respectively and incubated overnight at 4°C. The samples were then washed twice with reactivation buffer, followed by treatment with the Donkey AlexaFluor 555 anti-Rabbit (Abcam; ab150074) at 1:400 for 2 hr at room temperature. After two washes with reactivation buffer, the samples were mounted on a slide and visualized using the spinning-disk confocal microscopy (*Figure 2A and E*).

To visualize actin structures in *Figure 1C*, *Figure 1—figure supplement 1H and I*, *Figures 3C,D, E*, and *Figure 4—figure supplement 1A*, spheroplasts were permeabilized and fixed as described above, and treated with CF633-phalloidin (Biotium; #00046; dissolved in water) at 1:20 for 10 min. Samples were washed twice with reactivation buffer before visualization.

In *Figure 1—figure supplement 1A and B*, cells and permeabilized spheroplasts were fixed with 3.7% formaldehyde for 12 min at room temperature. The fixed cells were permeabilized with PBS containing 1% Triton X-100 for 1 min, washed twice with PBS, and then treated with Rhodamine-conjugated phalloidin (Life Technologies; R415) at 1:20. Similarly, fixed and permeabilized spheroplasts were treated with Rhodamine-conjugated phalloidin at 1:20 to visualize actin structures.

## Spinning-disk confocal microscopy

All images, except *Figure 1D* and *Figure 1—figure supplement 1C*, were captured using the Andor Revolution XD spinning disk confocal system, which was equipped with a Nikon ECLIPSE Ti inverted microscope, Nikon Plan Apo Lambda 100×/1.45NA oil immersion objective lens, a spinning-disk system (CSU-X1; Yokogawa), and an Andor iXon Ultra EMCCD camera. Images were acquired at the pixel size of 80 nm/pixel using the Andor IQ3 software. Three Laser lines at wavelengths of 488 nm, 561 nm, and 640 nm were used for excitation. All images were acquired with Z-step sizes of 0.2 µm, 0.3 µm or 0.5 µm, at varied interval times in individual time-lapse microscopy experiments.

## STORM

Super resolution microscopy (*Figure 1D* and *Figure 1—figure supplement 1C*) was performed using a custom-built TIRF widefield microscope with enhanced stability (http://wosmic.org). The STORM microscope has been tested to have an axial resolution (x-y direction) of 22–28 nm, done as previously described (*Nieuwenhuizen et al., 2013*). For sample preparation, permeabilized spheroplasts were fixed with 3.7% formaldehyde, and washed twice with reactivation buffer. Recombinant GST-LifeAct-GFP protein (1.3 µg/µl) was then added at 1:100 into the fixed permeabilized spheroplasts and incubated overnight at 4°C. The sample was then washed twice with reactivation buffer, and further treated with the AlexaFluor 647 anti-GFP antibody (Life Technologies; A-21447) at 1:200 for 4 hr at room temperature, followed by two washes with reactivation buffer. The sample was resuspended in imaging buffer (80 mM PIPES [pH 6.7]; 10 mM $MgCl_2$; 1 mM EGTA; 50% v/v glucose, 10% v/v b-mercapthoethanol). Prior to imaging with STORM, 0.1 µl of GOC mix (5 mg/ml glucose oxidase and 1 mg/ml catalase) was added to 9.9 µl of samples, and then sandwiched between two coverslips. The STORM images were reconstructed using GDSC SMLM plugins from the University of Sussex installed in Fiji (*Schindelin et al., 2012*). To measure the bundle or ring width from STORM images, a portion of a bundle or a ring was selected and straightened (Fiji/Edit/Selection/ Straighten) using the segmented line function of Fiji. A vertical line was drawn every 200 nm along the straighten line. The intensity of pixels along the vertical line was measured (Fiji/Analyze/Plot Profile). The width was measured by taking the Full-Width-Half-Maximum (FWHM) value after fitting the

vertical line profile to a Gaussian curve function (Fiji/Analyze/Tools/Curve Fitting). These processes were repeated using a macro written in Fiji.

## Image analysis

Images were analyzed using Fiji (*Schindelin et al., 2012*) and Imaris (Bitplane). The image stacks were projected along the Z-axis (sum intensity or maximum intensity) for analysis and for representation. The sum intensity projections were performed in *Figure 1—figure supplement 1G,H,I,J*, *Figure 2F*, and *Figure 3E*. The maximum intensity projections were performed in *Figure 1A,B,C,E*, *Figure 1—figure supplement 1A,B,D,E,F*, *Figure 2A,B*, *Figure 2—figure supplement 1A,C,D*, *Figure 3B,C*, *Figure 3—figure supplement 1A*, *Figure 4A,B,C*, and *Figure 4—figure supplement 1A,B*.

The background of all microscopy images, except *Figure 1D*, *Figure 1—figure supplement 1C*, and *Figures 4B,C*, and *Figure 4—figure supplement 1B*, was subtracted in Fiji (Fiji/Process/Subtract Background).

All time-lapse microscopy images, except *Figure 1—figure supplement 1G* and *Figure 4C*, were corrected for photo-bleaching in Fiji (Fiji/Image/Adjust/Bleach Correction).

Imaris was used to facilitate three-dimensional measurements and representation. The ring perimeters (*Figure 2E*, *Figure 4F*), the number of bundles (*Figure 2Ei,Eiii*, *Figure 3D*, *Figure 4D* and *Figure 4—figure supplement 1A*), the rate of bundle expulsion from the ring (*Figure 2D,2G*, *Figure 4G*), and the length of bundles (*Figure 2Eii,Eiv,Ev*, *Figure 3D*), were quantified after loading raw image stacks without prior processing.

To measure ring contraction rate in *Figure 4F,H* and *Figure 2—figure supplement 1B*, the ring perimeter was calculated as the total length of the ring skeleton using the Matlab function 'regionprops'. The procedure was then repeated for each time frame of the time-lapse series. The resulting data were fitted to a linear regression model using Matlab function 'fitlm'. The ring contraction rate was then measured as the slope of a fitted line.

All time-lapse videos were edited and saved in MP4 format with H.264 compression. Graphs were made with Prism 6 (GraphPad). The figures were arranged with Adobe Illustrator.

## Curvature analysis

To obtain the local curvature of Rlc1p-GFP rings in *Figure 4A,C,E*, and *Figure 4—figure supplement 1B*, a similar approach was developed as previously described (*Driscoll et al., 2012*). Briefly, raw Z stacks containing the actomyosin ring were de-noised with non-local means filter using a plugin installed in Fiji (plugin source: http://imagej.net/Non_Local_Means_Denoise). To facilitate shape measurement, the 3D stack was converted to 2D maximum intensity projection along the normal to the ring plane (Fiji/Image/Stacks/3D_project) in each time point. As the ring plane was often tilted to the X-Y plane, the projection angle was determined manually for each ring prior to the projection. The shape of the ring at each time point was extracted from 2D projections using segmentation via auto-thresholding by Otsu (Fiji/Image/Adjust/Thresholding) and subsequent skeletonization of the resulting mask (Fiji/Process/Binary/Skeletonize). The centres of skeleton pixels were defined as skeleton points.

Further processing of ring skeletons was done in Matlab. In order to measure the local curvature of the ring skeleton at the point-of-interest (POI), two skeleton points that were 10 pixel points (2 μm apart along the skeleton) away from the POI in clockwise and counter-clockwise directions were assigned. Next, a circle with radius R was fitted to these three points. The local curvature at the POI was then derived as the inverse of the radius of that circle (1/radius). For representation in figures, the curvature was smoothened over 10 skeleton points.

## Purification of the recombinant LifeAct-GFP fusion protein

Expression of GST-LifeAct-GFP was induced in *E. coli* BL21 (DE3-pLysS) (Promega) using 0.5 mM IPTG at 30°C for 4 hr. The recombinant protein was purified on Glutathione sepharose 4B beads according to the manufacture's instructions (GE Healthcare). The elution buffer containing glutathione was exchanged to reactivation buffer (0.16 M sucrose, 5 mM $MgCl_2$, 50 mM potassium acetate, 20 mM MOPS–NaOH [pH 7.0], pH adjusted to 7.5) using PD Minitrap G-10 columns (GE Healthcare). The purified recombinant proteins were stored at −80°C.

## Immunoblotting

The immunoblot in *Figure 3F* was prepared as follows: cell ghost pellet and supernatant were separated by centrifugation (15000 xg, 5 min at 4°C) after 0.5 mM ATP addition for 20 min and 40 min. One millilitre of TCA precipitation buffer (250 mM NaOH, 7.5% TCA) was added to resuspend the cell ghost pellet and supernatant, and the suspension was incubated on ice for 10 min. Samples were centrifuged at 17,000 xg for 25 min at 4°C. Precipitated proteins were resuspended in HU-DTT (200 mM Tris-HCl [pH 6.8], 8 M urea, 5% SDS, 0.1 mM EDTA, 0.005% bromophenol blue, and 15 mg/ml DTT) as previously described (*Palani et al., 2012*). Samples were heated for 5 min at 95°C before loading on SDS-PAGE gels. Antibodies used were goat anti-actin (generous gifts from John Cooper, Washington University School of Medicine, St Louis, USA) and rabbit anti-Cdc8 (*Balasubramanian et al., 1992*). Secondary antibodies used were rabbit anti–goat, goat anti–rabbit, IgGs coupled to horseradish peroxidase (NEB; Jackson ImmunoResearch Laboratories).

## Quantification of protein intensities

The signal intensities of indicated protein bands on immunoblots were measured using Fiji. Signal intensities were corrected against the gel background signal. The band intensity of the supernatant (S) and pellet (P) lanes was measured separately and summed up. To quantitate the fraction, the band intensity of supernatant lanes was divided by the sum of supernatant 'S' and pellet 'P' (fraction = band intensity/sum; sum = S + P). Four measurements were taken from blots derived from four independent experiments in *Figure 3F*.

## Statistical analysis

Statistical significance was determined using Student's t-test in *Figure 1D*, *Figure 1—figure supplement 1H,I*, *Figure 2Eii,Eiv*, *Figure 3E,F*, and *Figure 4H*. Calculations of mean, standard deviation (s.d.), and statistical significances, were done using Prism 6.0 (GraphPad).

# Acknowledgements

We would like to acknowledge members of the MKB. laboratory for discussion. This work was supported by Wellcome Trust Senior Investigator Awards to MKB. (WT101885MA), RAC. (103895/Z/14/Z), and SO (103741/Z/14/Z). MKB. was also supported by a Royal Society Wolfson Merit Award.

# Additional information

### Competing interests

MKB: Reviewing editor, *eLife*. The other authors declare that no competing interests exist.

### Funding

| Funder | Grant reference number | Author |
| --- | --- | --- |
| Wellcome Trust | 103895/Z/14/Z | Robert Anthony Cross |
| Wellcome Trust | 103741/Z/14/Z | Snezhana Oliferenko |
| Wellcome Trust | WT101885MA | Mohan K Balasubramanian |
| Royal Society | | Mohan K Balasubramanian |

The funders had no role in study design, data collection and interpretation, or the decision to submit the work for publication.

### Author contributions

JH, TGC, Conception and design, Acquisition of data, Analysis and interpretation of data, Drafting or revising the article; YG, Generated yeast strains and edited the manuscript, Contributed unpublished essential data or reagents; SP, AK, DSM, NJC, Acquisition of data, Analysis and interpretation of data; RAC, Acquisition of data, Analysis and interpretation of data, Drafting or revising the article; SO, Drafting or revising the article, Contributed unpublished essential data or reagents; MKB, Conception and design, Analysis and interpretation of data, Drafting or revising the article

## Author ORCIDs

Robert Anthony Cross, http://orcid.org/0000-0002-0004-7832

Mohan K Balasubramanian, http://orcid.org/0000-0002-1292-8602

## Additional files

**Supplementary files**

• Supplementary file 1. Table shows fission yeasts S. *japonicus* and S. *pombe* strains used in this study.

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
