## [Decision Letter]

[Editors’ note: a previous version of this study was rejected after peer review, but the authors submitted for reconsideration. The first decision letter after peer review is shown below.]

Thank you for submitting your work entitled "Curvature-Induced Expulsion of Actomyosin Bundles During Cytokinetic Ring Contraction" for consideration by *eLife*. Your article has been reviewed by two peer reviewers, and the evaluation has been overseen by a Reviewing Editor and Richard Losick as the Senior Editor. The following individuals involved in review of your submission have agreed to reveal their identity: Michael Glotzer (Reviewer #1).

Our decision has been reached after consultation between the reviewers. Based on these discussions and the individual reviews below, we regret to inform you that your work will not be considered further for publication in *eLife* at this point.

Although both reviewers and the reviewing editor found your work of interest, they were unable to come to a conclusion until your data was better quantified. In particular we agree that a major flaw is the lack of a quantitative analysis of the relationship of the extruded actin to the ring actin.

We do note that this is a report, describing the initial finding of an exciting new mechanism for ring disassembly. Therefore, while the primary conclusions must be rigorously established, I think that some of the less central points could be stated less strongly and strengthened in future publications.

Therefore, I would recommend that you follow the suggestions of the reviewers to further quantify your work. If you feel that this still supports your conclusions, *eLife* would be happy to see a new manuscript.

*Reviewer #1:*

This interesting report presents strong evidence that under some conditions, bundles of actin are expelled by the contractile ring during constriction, rather than being depolymerized in place. The expulsion of actin bundles is more prominent as constriction occurs, can be seen in cell ghosts, and mechanical perturbations reveal that this specifically occurs at regions of high local curvature. The images provided are striking and the data are well quantified. These findings will be of interest for the readership of *eLife*.

1) While the authors do demonstrate this phenomenon in a wide variety of contexts including two different *Schizosaccharomyces* species, different labels, intact cells, spheroplasts, cell ghosts, upon seeing these images it is striking that this was not seen previously in all the extensive work on *S. pombe* cytokinesis, particularly as investigators have been visualizing cytokinesis en face quite frequently in recent years. Indeed, en face views with a similar/identical label (Rlc1) does not appear to show this expulsion behavior (PMID 25355954). In addition, some, but not all the experiments in this paper have been performed with LifeAct which can induce filament bundling. To ensure that this phenomenon is general and robust and intrinsic to the contractile ring, it would be important to visualize evidence of this behavior in intact cells. It would be best to fix LifeAct-GFP, Rlc1-GFP, and unlabeled cells of both *S. pombe* and *S. japonicus* and stain them with phalloidin. It would be critical to detect these extruded filaments in cells that do not express a fluorescent protein (performing the experiment with the FP will allow the authors to assess that these bundles can be preserved and labeled).

*Reviewer #2:*

It is well-established that the cytokinetic ring closes via actomyosin dynamics. A more poorly understood aspect of ring ingression is that, as its diameter decreases, ring components are likely eliminated to allow continued contraction. In this manuscript, Huang et al. propose that actin and myosin containing filaments can be continuously expulsed from the contractile ring contributing to the completion of cytokinesis in two yeast species. Since the authors only observe such filaments in late stages of cytokinesis they suggest that this represents a separate phase during cytokinesis where filament expulsion promotes ring closure. Furthermore, the authors claim that this expulsion phenomenon is dependent on changes in cell curvature and is driven by mechanical forces generated by high curvature independent of cytoplasmic regulators or polymerization driven elongation of these filaments.

While the model proposed by the authors is intriguing and has the potential to expand our understanding of the mechanism of contractile ring closure, in its current form, the manuscript fails to support many of the authors' assertions and requires significant revisions and improvements before being suitable for publication in *eLife*.

In general, much of the data are poorly annotated in the figure, figure legend, text, or all three places. Some data also appear to be contradictory to the conclusion drawn by the authors or lacking quantification to support the conclusions. The following is a list of comments in chronological order. I cannot predict whether a revision will be suitable for publication before these points are addressed.

1) Many graphs presented in this manuscript contain the axis label relative intensity (AU) without any mention of what the intensity is normalized to. Typically, when relative values are shown in a graph, one of the data sets presented has the average of 1, and thus indicates that it is the reference point. This is not the case for any of the graphs (non-exhaustive list of figures affected: Figure 1—figure supplement 1, Figure 1—figure supplement 1, Figure 3, Figure 3). The notation of figures, in general, needs to be improved. "B.r. and s.r." are not appropriate abbreviations – writing out "big and small" would be adequate and far less cumbersome for the reader. The protein imaged should always be part of the image notation. Spectrum bars need to be annotated on the figure.

2) The data presented in the top and bottom panels of Figure 1—figure supplement 1 are not consistent. The bottom panel shows drastic reduction in fluorescence in the ring over time while levels of ring fluorescence in the top seems to change very little. Is the top series saturated? Which one is representative of properly imaged ring dynamics?

3) What is the difference between Figure 1—figure supplement 1? 1C shows a 3-fold decrease in fluorescence intensity while the intensities in Figure 1—figure supplement 1 are barely statistically significantly different. (How is total actin detected and distinguished from actin in the ring in Figure 1—figure supplement 1?) In fact, claiming that there is a significant difference between small and large rings in 1F is tenuous at best since the difference is minimal and could well be explained by a few outlying data points.

4) The following conclusion is not supported by the data presented: "Cdc8p, IQGAP protein Rng2p, and F-BAR protein Cdc15p were also present in actin bundles associated with contracting rings (Figure 1—figure supplement 1), which suggested that the bundles associated with contracting actomyosin rings signified disassembly of the entire ring as a single unit". All that is shown is that the bundles contain these components. Whether they have been part of a disassembling ring or just associated with filaments after expulsion is not clear. Along the same line, the authors observe bi-directional movement of myosin in expulsed bundles. Is this behavior observed in the ring? If not, wouldn't this suggest that these filaments are structurally distinct from filaments in the ring?

5) The biphasic nature of ring closure speed appears to be a minor, poorly substantiated observation, since it is entirely possible that expulsion starts to happen earlier but that the filaments are short and therefore not detectable. The really interesting observation in my opinion is the relationship between ring perimeter and bundle length. There appears to be a pretty good correlation in phase 2, suggesting that expulsion is a compensatory mechanism for reduction in ring size. However, the quantification of this in Figure 1 is confusing and contradictory. What are the units in the graphs? It appears that total length of bundles is larger than ring perimeter (max value 58 vs. 28) but T.I. + perimeter is smaller than perimeter alone? In addition, biphasic behavior of ring intensity (constant in phase 1, decreasing in phase 2), would further substantiate this biphasic concept.

6) The following statement is not supported by the data presented "These experiments showed that loss of function of actomyosin ring proteins tested did not affect the timing or the ability of these cells to expel actomyosin bundles during ring contraction". All that is shown are two pictures for each condition showing that small rings have bundles but no quantification is provided. All one can conclude from this is that neither of those proteins is essential for formation of these bundles. What about the kinetics, number of bundle, length or dynamics? Given the amount of redundancy in regulating cytokinesis, it would not be surprising if any one perturbation does not completely abolish the formation of bundles but the protein in question may still contribute to bundle formation and dynamics.

7) When making cell ghosts it appears that rlc1p-gfp is no longer associated with actin bundles. Is this just not evident from the picture shown or does this mean that myosin is lost from the bundle?

8) How is total actin intensity measured in cell ghost experiments? Just the sum of intensities on ring and filaments or total fluorescence in entire field of view? I presume Cdc8p is a loading control for western blots but this needs to be explicitly stated in the text. What is the quantification of the western blot comparing? Supernatant to pellet levels, loading control to actin monomers, filaments? It appears that most actin in the preparation is monomeric so that a significant reduction in filamentous actin may not be easily detected by changes in monomeric actin.

9) The authors claim that expulsion depends on ring curvature in non-compressed cells The data presented are insufficient to support this conclusion. All I see in Figure 4 is a picture. No quantitative data except for an arbitrary value of 68% of something, with no explanation of what is considered high curvature. In fact, based on the data presented it would be reasonable to conclude that high curvature is not essential for expulsion since a large fraction of filaments form in regions of "low" curvature (32%). In fact, maximum expulsion seems to occur at intermediate curvatures not at high curvature and small rings with high curvature appear to have fewer expulsion events.

10) The description of experimental details in the Materials and methods section needs significant revision. Just one example: "Unless specified otherwise, all spinning-disk images were shown by 2D maximum/sum intensity projections." Which one is it? There is a critical difference between sum and max projections. What do the authors mean by 2D projection? Do they mean Z-projection?

[Editors’ note: what now follows is the decision letter after the authors submitted for further consideration.]

Thank you for resubmitting your work entitled "Curvature-Induced Expulsion of Actomyosin Bundles During Cytokinetic Ring Contraction" for further consideration at *eLife*. Your revised article has been favorably evaluated by Richard Losick (Senior editor), a Reviewing editor, and one reviewer.

The authors have made significant changes to their manuscript that documents that during cytokinesis actin bundles with its associated binding proteins are expelled from the constricting contractile ring, particularly as it reaches small diameters. The authors make a compelling case that this is a common occurrence and that it constitutes a mechanism by which a significant portion of actin is eliminated from the ring.

But there are some remaining issues that need to be addressed before acceptance, as outlined below:

The data shown here, indicates that as the ring shrinks from a perimeter of ~26 µm to ~13 µm, relatively few bundles are observed, despite ~half of the f-actin being lost during this first period of constriction. There are three implications from this finding. First, it suggests that at early stages, "classical" disassembly is probably responsible for the loss of actin and ABPs. Second, unless ~1/3 of the actin is lost by extrusion during these early stages, the value of 68% of total ring lost by extrusion seems quite high (the solution to this appears to be that this value is calculated from rings that have already constricted significantly – a 3.5 µm diameter ring is not a typical early ring, other data in the manuscript indicates that an early ring is ~ 8 µm in diameter). Thus, this calculation appears somewhat misleading. Third, the manuscript should discuss how these two processes likely work together during ring constriction.

---

## [Author Response]

[Editors’ note: the author responses to the first round of peer review follow.]

Reviewer #1:

[…]

*1) While the authors do demonstrate this phenomenon in a wide variety of contexts including two different Schizosaccharomyces species, different labels, intact cells, spheroplasts, cell ghosts, upon seeing these images it is striking that this was not seen previously in all the extensive work on S. pombe cytokinesis, particularly as investigators have been visualizing cytokinesis en face quite frequently in recent years. Indeed, en face views with a similar/identical label (Rlc1) does not appear to show this expulsion behavior (PMID 25355954). In addition, some, but not all the experiments in this paper have been performed with LifeAct which can induce filament bundling. To ensure that this phenomenon is general and robust and intrinsic to the contractile ring, it would be important to visualize evidence of this behavior in intact cells. It would be best to fix LifeAct-GFP, Rlc1-GFP, and unlabeled cells of both S. pombe and S. japonicus and stain them with phalloidin. It would be critical to detect these extruded filaments in cells that do not express a fluorescent protein (performing the experiment with the FP will allow the authors to assess that these bundles can be preserved and labeled).*

We agree with reviewer 1 that bundle expulsion that we report here was not obvious in the Fred Chang lab paper (Zhou et al:PMID 25355954). However, recently (while our paper was under review / revision) a similar expulsion phenomenon was noticed in *S. pombe* cells in a paper from Daniel Riveline’s group (Wollrab et al:PMID 27363521), although they did not characterize this bundle expulsion phenomenon. Rather they focused on a different aspect of actomyosin ring dynamics.

As suggested by reviewer 1, we performed formaldehyde fixation of untagged wild type cells and spheroplasts and wild-type cells and spheroplasts expressing LifeAct- GFP or Rlc1p-GFP in both *S. japonicus* and *S. pombe* (to ensure that the ring- associated bundles observed in our study was not an artifact of LifeAct-GFP or Rlc1p-GFP). After staining the actin structures with Rhodamine-conjugated phalloidin in these fixed samples, we were able to observe actin bundles associated with contracting actomyosin rings, including in untagged wild type cells. We have included these images in Figure 1—figure supplement 1. This experiment clearly established that the bundles expelled during ring contraction were not an artifact of using non-native fluorescent markers, such as LifeAct-GFP or Rlc1p-GFP.

*Reviewer #2:*

[…]

*While the model proposed by the authors is intriguing and has the potential to expand our understanding of the mechanism of contractile ring closure, in its current form, the manuscript fails to support many of the authors' assertions and requires significant revisions and improvements before being suitable for publication in eLife.*

*In general, much of the data are poorly annotated in the figure, figure legend, text, or all three places. Some data also appear to be contradictory to the conclusion drawn by the authors or lacking quantification to support the conclusions. The following is a list of comments in chronological order. I cannot predict whether a revision will be suitable for publication before these points are addressed.*

We thank reviewer 2 for his/her detailed comments. We have carefully gone through and re-annotated the figures, and re-written/expanded figure legends, main text and Materials and methods. We believe these changes improve the clarity and accuracy of presentation.

*1) Many graphs presented in this manuscript contain the axis label relative intensity (AU) without any mention of what the intensity is normalized to. Typically, when relative values are shown in a graph, one of the data sets presented has the average of 1, and thus indicates that it is the reference point. This is not the case for any of the graphs (non-exhaustive list of figures affected: Figure 1—figure supplement 1, Figure 1—figure supplement 1, Figure 3, Figure 3). The notation of figures, in general, needs to be improved. "B.r. and s.r." are not appropriate abbreviations – writing out "big and small" would be adequate and far less cumbersome for the reader. The protein imaged should always be part of the image notation. Spectrum bars need to be annotated on the figure.*

We thank reviewer 2 for pointing out the incorrect axis label. This is unfortunately a mistake on our part in choosing the incorrect terminology. The referee is right in that these are not relative units, but they are the raw intensity of fluorescence of the proteins described. We have replaced all incorrect axis labels by showing the identity of the protein measured (e.g. actin intensity). All the micrographs are labeled with the protein being imaged and spectrum / calibration bars have also been included in the micrographs in the current manuscript. Abbreviations “b.r. and s.r.” have been replaced by “large and small”, as suggested.

*2) The data presented in the top and bottom panels of Figure 1—figure supplement 1 are not consistent. The bottom panel shows drastic reduction in fluorescence in the ring over time while levels of ring fluorescence in the top seems to change very little. Is the top series saturated? Which one is representative of properly imaged ring dynamics?*

The inconsistency of top and bottom panels in the previous Figure 1—figure supplement 1 was due to an unfortunate and erroneous swap of the bleach corrected panel with a bleach-uncorrected panel (we generated both versions during preparation of the manuscript). We now show the appropriate bleach corrected micrographs in the top and bottom panels that reveals the phenomenon of bundle expulsion even better than in the original manuscript (shown in Figure 1—figure supplement 1 in the revised manuscript).

*3) What is the difference between Figure 1—figure supplement 1? 1C shows a 3-fold decrease in fluorescence intensity while the intensities in Figure 1—figure supplement 1 are barely statistically significantly different. (How is total actin detected and distinguished from actin in the ring in Figure 1—figure supplement 1?) In fact, claiming that there is a significant difference between small and large rings in 1F is tenuous at best since the difference is minimal and could well be explained by a few outlying data points.*

The previous Figure 1—figure supplement 1 showed the intensity of actin present in the ring (excluding ring-associated actin bundles) in the analysis. The previous Figure 1—figure supplement 1 showed the combined actin intensity (total actin) from rings and associated bundles. Please note that the Figure 1—figure supplement 1 and 1F are now Figure 1—figure supplement 1 and 1I in the current manuscript.

To measure the sum actin fluorescence intensity, the permeabilized spheroplasts were stained with CF633-phalloidin to label the actin structures. The total actin was measured by drawing a square to cover the actomyosin ring and the associated bundles as the region-of-interest, whereas the ring associated actin was measured by drawing a line along the ring circumference without including ring-associated bundles as the region-of-interest. These details are provided in the corresponding figure legend.

In the previous Figure 1—figure supplement 1 we were not claiming any difference between total actin in large and small rings. Rather, based on the statistics presented, we were claiming that they were of comparable intensity (as the referee agrees from his/her own analysis). Our apologies that this was described confusingly. This has been clearly rewritten now.

*4) The following conclusion is not supported by the data presented: "Cdc8p, IQGAP protein Rng2p, and F-BAR protein Cdc15p were also present in actin bundles associated with contracting rings (Figure 1—figure supplement 1), which suggested that the bundles associated with contracting actomyosin rings signified disassembly of the entire ring as a single unit". All that is shown is that the bundles contain these components. Whether they have been part of a disassembling ring or just associated with filaments after expulsion is not clear. Along the same line, the authors observe bi-directional movement of myosin in expulsed bundles. Is this behavior observed in the ring? If not, wouldn't this suggest that these filaments are structurally distinct from filaments in the ring?*

We have performed time-lapse microscopy to document the dynamics of two cytokinetic ring proteins: GFP-Rng2p and Cdc15p-GFP (in Figure 2), in addition to Rlc1p-GFP, which was presented in the original submission. Consistent with our original description from single images, these proteins were found to be associated with the expelling / expelled bundle in contracting rings using time-lapse microscopy (presented in the revised manuscript, Figure 2). Despite this, we have toned down the statement (for reasons described in the response to #7 below) that “the bundles associated with contracting actomyosin rings signified disassembly of the entire ring as a single unit”, by stating that several cytokinetic ring proteins associate with expelling / expelled bundles in living cells and discussed this in the manuscript (Results and Discussion, tenth paragraph).

We have removed the data showing bi-directional movements of Rlc1p-GFP on expelled bundles as, in our opinion, it is less central to our study describing bundle expulsion. This is partly also due to the suggestion from the editors to remove data not central to the main point of our manuscript. However, we have been able to detect clusters moving in the ring, although this point has been better described in a recent paper from Daniel Riveline and colleagues that was published (Woolrab et al; PMID 27363521) while our paper was under review/revision.

*5) The biphasic nature of ring closure speed appears to be a minor, poorly substantiated observation, since it is entirely possible that expulsion starts to happen earlier but that the filaments are short and therefore not detectable. The really interesting observation in my opinion is the relationship between ring perimeter and bundle length. There appears to be a pretty good correlation in phase 2, suggesting that expulsion is a compensatory mechanism for reduction in ring size. However, the quantification of this in Figure 1 is confusing and contradictory. What are the units in the graphs? It appears that total length of bundles is larger than ring perimeter (max value 58 vs. 28) but T.I. + perimeter is smaller than perimeter alone? In addition, biphasic behavior of ring intensity (constant in phase 1, decreasing in phase 2), would further substantiate this biphasic concept.*

We have removed the description of the biphasic ring contraction behaviour in the current manuscript, again to streamline and focus on the major point that ring disassembly occurs in large part through bundle expulsion. The graphs have been updated with the appropriate units (Figure 2). We show individual graphs in Figure 2 without insets so that data points are clearer.

Thanks to Reviewer 2 for pointing out the confusing data points, i.e. total bundle length + ring perimeter in contracted rings is smaller than the initial ring perimeter. We investigated the graph again, and realized that the upper limit for y-axis (inset of 3^rd^ graph in our previously submitted manuscript) was unfortunately set at a lower value erroneously, which eliminated some of the important points. We have rectified this mistake and the revised graph fully supports the conclusions that we have made (Figure 3Ev).

*6) The following statement is not supported by the data presented "These experiments showed that loss of function of actomyosin ring proteins tested did not affect the timing or the ability of these cells to expel actomyosin bundles during ring contraction". All that is shown are two pictures for each condition showing that small rings have bundles but no quantification is provided. All one can conclude from this is that neither of those proteins is essential for formation of these bundles. What about the kinetics, number of bundle, length or dynamics? Given the amount of redundancy in regulating cytokinesis, it would not be surprising if any one perturbation does not completely abolish the formation of bundles but the protein in question may still contribute to bundle formation and dynamics.*

We have removed this section on the analysis of mutants, as these experiments pertain to *S. pombe*, whereas the vast majority of the study is on *S. japonicus*. Also, as pointed out by the referee, while our data are suggestive they are not conclusive, which requires generation of a large number of double and triple mutants in *S. pombe*, and crucially generation of many single mutants in *S. japonicus*, all of which are beyond the scope of this study.

*7) When making cell ghosts it appears that rlc1p-gfp is no longer associated with actin bundles. Is this just not evident from the picture shown or does this mean that myosin is lost from the bundle?*

We have been able to detect Rlc1p, Cdc15p, Rng2p, and F-actin in expelled bundles in cells and spheroplasts, but only F-actin was clearly detected in expelled bundles in cell ghosts. Although the reasons for this are presently unclear, it is likely that ring proteins undergo constant turnover in expelled actin bundles (as they do in actomyosin rings) in cells and spheroplasts in which a continuous supply of a molecularly crowded cytosol is available, whereas this is not available in cell ghosts. Alternatively, the molecular crowding itself may help retain ring proteins on actin bundles whereas in the absence of such crowding, ring proteins may be lost from actin bundles. We have discussed this in the revised manuscript (Results and Discussion, eleventh paragraph).

*8) How is total actin intensity measured in cell ghost experiments? Just the sum of intensities on ring and filaments or total fluorescence in entire field of view? I presume Cdc8p is a loading control for western blots but this needs to be explicitly stated in the text. What is the quantification of the western blot comparing? Supernatant to pellet levels, loading control to actin monomers, filaments? It appears that most actin in the preparation is monomeric so that a significant reduction in filamentous actin may not be easily detected by changes in monomeric actin.*

To measure the sum actin fluorescence intensity in cell ghosts, they were stained with CF633-phalloidin to label the actin structures. The total actin intensity was measured by drawing a square to cover the actomyosin ring and the associated bundles as the region-of-interest, whereas the ring associated actin was measured by drawing a line along the ring circumference without including ring-associated bundles as the region-of-interest. These details are provided in the corresponding figure legend.

The western blots compare actin and an actin-associated protein (Cdc8p- tropomyosin) in the supernatant and pellet. Cdc8p was not used as a loading control. We have changed the axis labeling to “supernatant fraction”. This fraction is calculated by dividing the protein band intensity of supernatant lanes to the sum intensity of the bands of supernatant lanes and pellet lanes (fraction = supernatant intensity / (supernatant intensity + pellet intensity). We have expanded the descriptions of protein band intensity quantification in the Materials and methods section of the current manuscript.

*9) The authors claim that expulsion depends on ring curvature in non-compressed cells The data presented are insufficient to support this conclusion. All I see in Figure 4 is a picture. No quantitative data except for an arbitrary value of 68% of something, with no explanation of what is considered high curvature. In fact, based on the data presented it would be reasonable to conclude that high curvature is not essential for expulsion since a large fraction of filaments form in regions of "low" curvature (32%). In fact, maximum expulsion seems to occur at intermediate curvatures not at high curvature and small rings with high curvature appear to have fewer expulsion events.*

The data that support the conclusion that expulsion depends on ring curvature was/is presented in Figure 4 in the original and revised manuscript. In the representative movie shown (n = 11) in Figure 4, expelled bundles become visible from ~ the 60-second time point, and with progression of time and increasing curvature the expelled bundle number increases. We find that although rings can start with curvatures (1/radius) of even smaller than 0.4 µm_-1_ (translates to a ring diameter of > 5 µm in spheroplasts), the mean curvature at which bundles begin to be expelled is ~ 1.1 µm_-1_ (translates to a ring diameter of ~ 2 µm). Given that bundle expulsion is only observed after a significant reduction in ring diameter, we believe our data strongly support the view that bundle expulsion in uncompressed rings is promoted by increased curvature. We have then tested this hypothesis with the spheroplast compression experiment.

Our apologies for the confusing description of Figure 4—figure supplement 1. In this panel we are looking at a partially contracted ring in a cell ghost. Rings in cell ghosts do not undergo symmetric contraction. Rather, they undergo irregular contraction with regions of sharp edges. What we were indicating in this figure was that, even within a curved and contracted ring in a cell ghost ~ 68% of actin bundles are expelled from regions where the ring is bent. In other words, *the entire ring has an increased curvature*, due to partial contraction, but nearly 68% of the cables are found preferentially in regions in which the rings are bent, which means that the rings have a further increased local curvature. We were unable to perform curvature analysis on contracting isolated rings in cell ghosts because the rings do not contract on a flat x-y plane. The poor z-resolution affects the accuracy of curvature analysis. We took precaution not to over-estimate the number, thus, the 68% is a conservative value of this quantification.

Together, Figure 4 and Figure 4—figure supplement 1, establish that bundle expulsion is promoted by increased ring curvature even in uncompressed cells.

*10) The description of experimental details in the Materials and methods section needs significant revision. Just one example: "Unless specified otherwise, all spinning-disk images were shown by 2D maximum/sum intensity projections." Which one is it? There is a critical difference between sum and max projections. What do the authors mean by 2D projection? Do they mean Z-projection?*

We fully appreciate that maximum intensity and sum intensity projections are very different. Our apologies for not making this point clear in the original manuscript. This has been fully rectified in the revised manuscript. We have clearly stated whether an image shows a maximum intensity or sum intensity in the Methods section, which clarifies the panels in which maximum intensity was used and those in which sum intensity was used. The referee is right in that we are using Z- projection to show a 2-dimensional image in the X-Y plane. This has now been corrected to “projected along the Z-axis”.

[Editors' note: the author responses to the re-review follow.]

*The authors have made significant changes to their manuscript that documents that during cytokinesis actin bundles with its associated binding proteins are expelled from the constricting contractile ring, particularly as it reaches small diameters. The authors make a compelling case that this is a common occurrence and that it constitutes a mechanism by which a significant portion of actin is eliminated from the ring.*

*But there are some remaining issues that need to be addressed before acceptance, as outlined below:*

The data shown here, indicates that as the ring shrinks from a perimeter of ~26 µm to ~13 µm, relatively few bundles are observed, despite ~half of the f-actin being lost during this first period of constriction. There are three implications from this finding. First, it suggests that at early stages, "classical" disassembly is probably responsible for the loss of actin and ABPs. Second, unless ~1/3 of the actin is lost by extrusion during these early stages, the value of 68% of total ring lost by extrusion seems quite high (the solution to this appears to be that this value is calculated from rings that have already constricted significantly – a 3.5 µm diameter ring is not a typical early ring, other data in the manuscript indicates that an early ring is ~ 8 µm in diameter). Thus, this calculation appears somewhat misleading. Third, the manuscript should discuss how these two processes likely work together during ring constriction.

We have read this comment and our manuscript carefully, and do not understand how the reviewer arrived at ~ half of the F-actin being lost during this first period of constriction. We chose early rings as those with a diameter of ~ 3.5 µm since in other experiments in Figure 2, we only observed bundles when rings reached a diameter of ~ 2.5 µm. Since the goal of the experiment suggested by reviewer 1 was to estimate the fraction of the ring-associated actin lost through bundle expulsion, an initial diameter of ~ 3.5 µm seemed like a good starting point. We therefore do not believe the reported calculations are misleading. As to the issue of whether actin is also lost through other “classical” disassembly pathways, this is an interesting and open question. Further analysis, using better probes for the actin cytoskeleton will be required to fully address this possibility.

As suggested by this reviewer, we have made a mention of these considerations in the revised manuscript.